# Fostering Animal Welfare and Advancing 3Rs Principles through the Establishment of a 3Rs Advisory Group

**DOI:** 10.3390/ani13243863

**Published:** 2023-12-15

**Authors:** Jessica C. Graham, Lisa Wong, Adeyemi O. Adedeji, Aija Kusi, Becky Lee, Donna Lee, Noel Dybdal

**Affiliations:** 1Product Quality and Occupational Toxicology, Genentech, Inc., South San Francisco, CA 94080, USA; 2Non-Clinical Operations, Genentech, Inc., South San Francisco, CA 94080, USA; lisawong@gene.com (L.W.); mckenza6@gene.com (A.K.); 3Safety Assessment Pathology, Genentech, Inc., South San Francisco, CA 94080, USA; adedejia@gene.com (A.O.A.); leeb53@gene.com (B.L.); nod@gene.com (N.D.); 4Safety Assessment Toxicology, Genentech, Inc., South San Francisco, CA 94080, USA; donnawl@gene.com

**Keywords:** 3Rs, animal welfare, internal 3Rs advisory group

## Abstract

**Simple Summary:**

Based on the current state of science, the use of animals remains essential in bringing safe and effective medicines to patients. To be on the forefront of science from an animal welfare and 3Rs (reduction, refinement, replacement) perspective, organizations benefit from an internal 3Rs advisory group. The goals of a 3Rs AG are to foster awareness and support the promotion of a culture of care (for humans and animals), drive innovation, accelerate technical development, and influence 3Rs best practices via a science-driven and collaborative approach. A 3Rs AG is a benefit to a company, enabling and supporting a pathway enriched with best practices, maintaining employee engagement and encouraging innovation. The thoughtful reduction in and replacement of animal studies and the refinement of animal study designs, enabled by a 3Rs AG, can enhance animal welfare and guide resources. This article provides guidance on how to establish a successful and impactful 3Rs AG.

**Abstract:**

Based on the current state of science, the use of animals remains essential in bringing safe and effective medicines to patients. Respect for laboratory animal welfare and the application of 3Rs principles (the replacement, reduction, and refinement of animal use in research) are a priority throughout the pharmaceutical industry. Given the rapid pace of development, technological progress, and the emergence of new-approach methodologies (NAMs) in the field of biomedical research, maintaining a leading position in scientific advancements with a focus on the principles of replace, reduce, and refine (3Rs) can be quite challenging. To effectively address these challenges and sustain a prominent position in the scientific community, organizations can derive significant advantages from establishing an internal 3Rs advisory group (3Rs AG). The primary objective of a 3Rs AG is to stay at the forefront of the knowledge of best practices related to the 3Rs principles in the industry. This group plays a crucial role in fostering innovation and facilitating the seamless integration and implementation of 3Rs principles into a company’s policies and procedures. The thoughtful reduction in and replacement of animal studies and the refinement of study designs and practices, enabled by a 3Rs AG, can minimize animal use as well as guide resources and positively impact study and data quality. This article provides guidance on how to establish a successful and impactful 3Rs AG.

## 1. Introduction

Despite important advances in alternative approaches, the use of animals remains a requirement in the preclinical evaluation of the potential hazards and safety risks of pharmaceuticals intended for human and animal patients alike [1]. Regulatory statutes globally recognize the importance of the humane care and use of laboratory animals in biomedical research, and these guidance vary from country to country. For example, in the United States, the federal “Animal Welfare Act” and “The Guide to the Care and Use of Laboratory Animals” provide regulations, while in the European Union, ETS 123 provides for the protection of vertebrate animals [2,3,4]. Such regulations set the foundation on which the humane treatment of laboratory animals is based. Additionally, as science advances, the understanding of species needs as well as societal expectations regarding what constitutes good animal welfare is also evolving. Russell and Burch (1959) acknowledged “the humanist possible treatment of experimental animals, far from being an obstacle to biomedical research, is actually a prerequisite for successful animal experiments” and progressed the principles of replacement, reduction, and refinement (3Rs) as a framework for animal use in research [5]. These principles serve as a guide, forming the basis for innovation and progress in animal welfare and scientific quality. This includes the continuous dedication of resources to support animal physical and psychological well-being, advancing beyond the basic ethical and moral obligations to reduce their suffering (e.g., federal regulations and official guidelines) [6].

The benefits of “a positive animal welfare state” are evident in relationship to scientific quality, extending beyond animal welfare and use reduction [5,7]. Consideration of the 3Rs (replacement of animal models, reduction in animal use, and refinement of animal studies to improve animal welfare) at every stage of drug development optimizes approaches and is a driving force in medical, scientific, and technological advancement [8]. Through the replacement of animals with alternative models and approaches, more targeted scientific questions and higher throughput data can be generated [9]. It also enables strategic resource management which is in the best interest of research organizations. For example, where in vivo studies are necessary, strategic statistical design, as well as housing and husbandry practices that support positive animal welfare states, can yield robust, reproducible data facilitating better decision making [9].

To ensure an alignment with 3Rs principles and best practices, animal welfare and use must be continuously strategized and scrutinized by all whose responsibilities include the physical and psychological health of research animals (e.g., researchers, contract research organizations, and regulatory agencies). Additionally, there is an increased probability of success, greater reach, and improved ease of implementation when experts work collectively toward a common goal to optimize 3Rs impact and drive change.

### Overview of a 3Rs Advisory Group (AG)

How does a company ensure it stays current with advances in the best practices related to 3Rs principles and animal welfare? The effective and efficient implementation of 3Rs principles within an organization is not without its challenges and thus highly benefits from an internal 3Rs advisory group (3Rs AG). For example, the “reduction of animal use” principle in particular has been criticized as leading to underpowered studies that are difficult to replicate. Additionally, the awareness of available alternative models and their acceptability by regulatory authorities can be uncertain. A 3Rs AG is an intentional and dedicated internal group of subject-matter experts (SMEs) who are attuned to developments in the field of animal welfare as well as advancements in study methodologies. A 3Rs AG monitors laboratory animal-welfare best practices as well as developments in 3Rs-related best practices and therefore can guide project teams with study methodology options that incorporate these principles while enabling desired and relevant high-quality data. A 3Rs AG provides a forum for individuals from varying backgrounds to discuss opportunities, learnings, best practices, and advancements across the industry.

## 2. Establishing a 3Rs AG

Key components in establishing and maintaining a thriving and impactful 3Rs AG include company sponsorship, effective leadership, dedicated members, and consistent organizational and cross-functional engagement (Figure 1). Roles and responsibilities can vary based on company size, focus, and capabilities (Table 1). For example, some companies may have an animal-welfare function, while others may have a fully grass-roots team which engages in supporting 3Rs principles across the organization and is fully responsible for its own success.

### 2.1. 3Rs AG Mission and Vision

The mission of a 3Rs AG is to support humane animal research by providing education on the advancement of methods to reduce, replace, and refine the use of animals in research. In a time where the demand for alternatives leading to improved scientific methodology and translation throughout the drug research and development process is high, there is a need for creative and collaborative approaches to maximize impact. A 3Rs AG also seeks to build a centralized resource for researchers to share emerging technologies that aid in the clinical translation of nonclinical in vivo models and NAMs. Therefore, the 3Rs AG is tasked with being aware of and addressing company needs where animal care and use are concerned as well as where opportunities exist to move the company and industry forward. Core focus areas for an effective 3Rs AG include awareness, influence, and innovation (Figure 2). Additionally, having a 3Rs AG charter that outlines sponsorship, leadership, meetings, objectives, and deliverables is ideal.


**An example mission and vision:**


The use of animals is critical to the development of safe and effective new medicines. Importantly, *company name* is committed to the ethical use of animals in research, a mission that is supported by the *company name*’s 3Rs advisory group (3Rs AG) which aims to identify and implement best practices across the organization.

The 3Rs AG mission is to support the 3Rs principles of the reduction, refinement, and replacement of animals in nonclinical studies of *company name*’s molecules, as well as support and accelerate technical developments to advance 3Rs principles within the industry.

The 3Rs AG vision is to continue to increase the awareness of 3Rs principles, drive innovation, and influence best practices to reduce animal use and enhance animal welfare across all nonclinical studies conducted within *company name*.

### 2.2. Organizational Support and 3Rs Leadership

In order to enable a successful 3Rs AG, accountability is necessary on the part of the company, 3Rs AG leadership, and membership. Company sponsorship extends the reach of the 3Rs AG, enabling its success and influence. Company support includes the recognition of 3Rs efforts (e.g., projects, awards) in company communications, funding to support various 3Rs-related activities (e.g., awards), and corporate leadership having a presence at 3Rs outreach activities. Having two co-leads in addition to a corporate sponsor is an ideal 3Rs AG leadership team as this affords flexibility and assistance with planning and implementing activities. A corporate sponsor can offer support for the 3Rs AG as well as be an advocate for the organization to aid in the adoption of 3Rs AG objectives. Additionally, administrative support can be helpful in ensuring that the dissemination of communications and planning of meeting spaces and activities are in place for the AG to function. Depending on the local culture, having 3Rs-AG-branded items (e.g., t-shirts, water bottles) can increase the visibility of a company’s commitment to animal welfare and encourage a sense of pride in 3Rs AG membership. Despite all these commitments, there is generally a low financial obligation required.

### 2.3. Cross-Functional Membership

Working effectively with individuals across functions is integral to the success of an aligned, innovative, and highly collaborative 3Rs AG. In forming a 3Rs AG, it is important to draw members from across the organization and to have cross-functional representation and expertise (Figure 3). Core members should be those who are closely involved with the planning and execution of animal studies. This core group is further enhanced by SMEs in biostatistics, in vitro/in silico technologies, nonclinical outsourcing, and translational research, to name a few. The inclusion of diverse areas of expertise not only enhances the 3Rs AG knowledge base and awareness of best practices and new technologies for in vivo studies but also facilitates learning and innovation through practical applications of 3Rs techniques within current projects (for example, see Section 4). This integration of functions can enable the identification of and focus on various in vivo, in vitro, and computational capabilities across an organization, increasing transparency and encouraging collaboration. A 3Rs AG with a diverse membership can also provide cross-functional oversight of in vivo studies, ensuring the integration of 3Rs principles from planning through to completion.

## 3. 3Rs AG Goals and Deliverables

Dedication and commitment from 3Rs AG leaders and members is inherent to a successful AG. Example routine and extended activities that support the mission of the 3Rs AG are presented in Table 2.

### 3.1. Internal Activities

3Rs AG meetings: Ideally, the whole 3Rs AG meets regularly (e.g., monthly) to share/discuss learnings and knowledge of cross-industry events, as well as challenges and practices related to animal welfare and the 3Rs. Initial meetings benefit from a focus on evaluating needs and challenges in the industry and company, establishing a mission and vision for the AG and discussing the AG’s scope. Ongoing meetings can focus on the refinement, reduction, and replacement for both improved animal welfare and scientific impact as part of short- and long-term goals. Additional topics include opportunities to influence, implement, and support strategies to enhance animal welfare for studies conducted internally as well as externally (i.e., at contract research organizations (CROs)). 

Communications: Established channels of communication are an essential element of an impactful 3Rs AG. Effective communication outlets provide information on global news and policies, upcoming events, recent publications, 3Rs AG activities, and links to relevant external organizations. Additionally, highlighting internal efforts and accomplishments in implementing and advancing 3Rs practices and animal welfare can add to a sense of employee pride and provide transparency regarding the company’s ethical and moral values as well as its investment in animal welfare. Examples of recommended communications include:An internal website to highlight the mission, vision, and goals of the 3Rs AG as well as provide a place to make relevant information available (e.g., regulations, guidance, publications, newsletters);A newsletter to disseminate information on recent and upcoming events and activities of interest to the 3Rs community and broader organization;Email communications as needed (yet minimized) to highlight current events, accomplishments, publications, webinars, etc., where more rapid communication can be impactful and drive current projects;Seminars on company accomplishments related to 3Rs principles, new technologies, and key areas of interest.

3Rs Awards: A 3Rs award program calls attention to the 3Rs as a company goal, drives innovation, and supports the integration of 3Rs principles into company practices. Ideally, these awards recognize teams engaged in activities (e.g., research, laboratory animal care) that support or advance 3Rs principles. These awards visibly acknowledge excellence in science as well as innovation and a positive animal-welfare culture.

3Rs Awareness Events and Outreach: A 3Rs awareness day (or week) can be a global event that includes seminars, lightning talks, and posters highlighting novel in vivo, in silico, and in vitro techniques and technologies of interest, continuous improvement in action, and the company’s overarching commitment to a positive animal-welfare state. These events can also include outreach activities where 3Rs AG members engage colleagues through displays highlighting company accomplishments (e.g., in a high-traffic area such as the cafeteria), distribute 3Rs-AG-branded goods, and facilitate contests where colleagues are prompted to think about the 3Rs. Similar to the 3Rs awards, these educational outreach activities celebrate and honor efforts to advance the 3Rs and promote awareness of ongoing activities and accomplishments across the organization and industry. They also serve as an additional advertisement for the AG and the company’s commitment to responsible animal use and animal welfare in general.

Recognition of Laboratory Staff: Integral to the culture of care, it is critical to convey company support to laboratory-animal personnel in order to nurture empathetic behavior and mitigate compassion fatigue [10]. An impactful 3Rs AG recognizes and celebrates the invaluable contributions of laboratory-animal technicians (e.g., AALAS’s Tech Week) as well as laboratory staff engaged in alternative methods. Such continual outreach and inclusion facilitates a positive company culture and can stimulate the awareness of opportunities for collaboration and innovation [11].

### 3.2. External Activities

In addition to guiding and positively impacting internal practices and policies, a 3Rs AG also strives to advance 3Rs practices and policies external to the organization. A key goal of a successful 3Rs AG is to influence and be at the forefront of animal-welfare policies and have a strong external scientific presence. Awareness of and influence on the external 3Rs environment can be accomplished through involvement with and the leadership of global/regional consortia and working groups, collaborations on cross-company projects and publications (e.g., exploring a new technique or approach with a CRO), and the organization of workshops/sessions and presentations at professional meetings. External engagement supports and enables 3Rs best practices within the organization as well as the advancement of innovations across the industry via science-driven and collaborative approaches.

Consortia with efforts focused on advancing the 3Rs in the pharmaceutical industry include the International Consortium for Innovation and Quality in Pharmaceutical Development (IQ), the European Partnership for Alternative Approaches to Animal Testing (EPAA), the New Jersey Association for Biomedical Research (NJABR), the National Centre for the Replacement, Refinement and Reduction of Animals in Research (NC3R^s^), and the Enhancing Translational Safety Assessment through Integrative Knowledge Management project (eTRANSAFE), to name a few.

### 3.3. Sources of Guidance for a 3Rs AG

There is a wealth of information available to guide a 3Rs AG in establishing and implementing practices that integrate 3Rs principles as well as contribute to a positive animal-welfare state. Several directives and documents that convey guidance on animal-welfare best practices are presented in Table 3. Additionally, many organizations offer perspectives and current news with a focus on 3Rs principles and offer guidance that can be utilized to fill in any gaps from an animal-welfare perspective (Table 4).

## 4. Case Studies of an Impactful 3Rs AG

Since its inception, the Genentech 3Rs AG has influenced and encouraged the integration of company best practices and strategies with 3Rs principles. Many accomplishments revolve around a reduction in animals whenever possible and a focus on animal welfare from the inception of nonclinical studies. Some of these initiatives are presented below and can be utilized to guide and inspire companies not yet implementing such practices.

Note that careful consideration of the principles of the 3Rs is necessary in determining the need for an animal study as well as in subsequent study design [12].

### 4.1. Reduction: Recovery Animals

**Inclusion of recovery animals is no longer a default approach for toxicology studies:** The main reason for the inclusion of recovery animals in a toxicology study is to assess the reversibility of any noted pathology changes. Over time, the inclusion of recovery animals has become a default approach rather than a case-by-case consideration. Through engaging with safety-assessment stakeholders (pathologists and toxicologists), it has become standard practice to make paper-based scientific assessments of reversibility in our base-case study design. Additionally, this base-case study design can be further modified to remove the mid-dose when adverse findings are not anticipated based on previous toxicity studies (Table 5). The 3Rs AG also recommended that recovery animals can be added based on factors highlighted in the recently published manuscripts [13,14].

### 4.2. Reduction: Control Animals

**Exclusion of a control group is our base-case study design for non-GLP (Good Laboratory Practices), dose-range-finding (DRF) studies:** The primary objective of a non-GLP DRF toxicology study is to assess tolerability and identify the maximum tolerated dose (MTD) of a new molecular entity (NME) to help inform dosing for a GLP and relatively longer-term study. To make these assessments, abnormal clinical signs of severe magnitude and mortality are considered the most important factors. Given that control animals are not needed to make these assessments, the 3Rs AG recommended that concurrent control groups should only be included in DRF studies for rodents or non-rodents with appropriate justification. This includes (a) using a novel vehicle, technology, or modality with no previous experience and no historical control data and (b) assessing for pharmacodynamic endpoints, cytokines, and other novel endpoints with no previous experience.

### 4.3. Refinement and Reduction: Small Plasma Sampling

**Elimination of toxicokinetic (TK) satellite animals for most rat toxicology studies:** The 3Rs AG have also made meaningful impact regarding the improvement of animal welfare with a concomitant reduction in animal numbers in our toxicology studies without negatively impacting the quality of the data. This effort involved reducing blood volume requirements for TK exposure assessments in rat toxicology studies. Given the improved small-molecule bioanalytical assay sensitivity and related decreased sample volume requirements for TK assessments, TK data collection in rat toxicity studies was reconsidered. Harstad et al. (2017) reported that utilizing small plasma sampling (i.e., 200 µL) in a sparse sampling study design (i.e., sample main study rats either once or twice for TK) reduced the need for satellite TK animals and allowed TK blood samples to be drawn from the main study toxicology animals, resulting in a 33% reduction in the animals needed [15].

## 5. Benefits of a 3Rs AG to the Company, Employees, and Industry

An internal 3Rs AG offers a multitude of benefits to an organization including fostering innovation, providing a forum for discussion of 3Rs-related topics, and supporting a positive company culture. A 3Rs AG can serve as a resource to company leadership, via providing input and feedback on emerging regulatory and guidance documents and through influencing policies and best practices across the industry. A 3Rs AG also aims to promote awareness as well as a culture of care through continued education via internal forums, newsletters, and guidance. Additionally, a 3Rs AG can serve as a hub for innovation and collaboration in strategizing and advancing study design and conduct, for example, enabling the cross-functional oversight of and collaboration on the implementation of new-approach methodologies (NAMs) (in vitro, in silico, in chemico).

A 3Rs AG can provide insight into whether an alternative model (in vitro or in silico) or a tiered approach (refining certain study aspects in order to use fewer animals) can be utilized, without compromising study quality. A 3Rs AG can also provide guidance on the best practices in animal handling, housing, and in vivo study conduct. A 3Rs AG increases employee engagement and serves as a reminder to the extended company that the organization has a rooted commitment to animal welfare. Overall, a 3Rs AG benefits employees (e.g., pride in their work, reduction in compassion fatigue), research animals (e.g., improved welfare), and the industry (e.g., forward-thinking innovations and improvements, positive perception, and value-added resource allocation).

## 6. Conclusions

The process of ensuring adherence to 3Rs principles and optimal animal-welfare practices involves collaboration across functions. Establishing a 3Rs AG leverages the scientific strengths of individual functions and involves initiatives led by cross-functional teams of volunteers within the organization to advance the 3Rs as a forefront of best scientific practices. By fostering information sharing, a 3Rs AG can discover alternative methods to animal use (such as in vitro and computational methods) and assess barriers to their implementation. Such continuous efforts to refine animal care and husbandry support a positive welfare state and high-quality data generation. A 3Rs AG is a benefit to a company, facilitating and supporting a pathway enriched with best practices, sustaining employee engagement, and fostering innovation.

## Figures and Tables

**Figure 1 animals-13-03863-f001:**
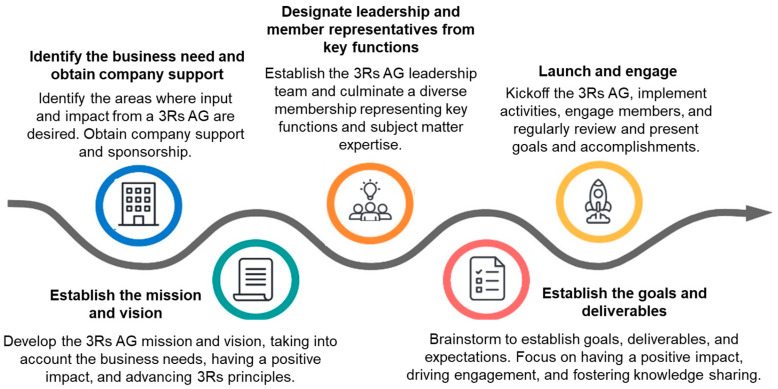
Establishing a 3Rs advisory group (3Rs—replacement, reduction and refinement, AG—advisory group).

**Figure 2 animals-13-03863-f002:**
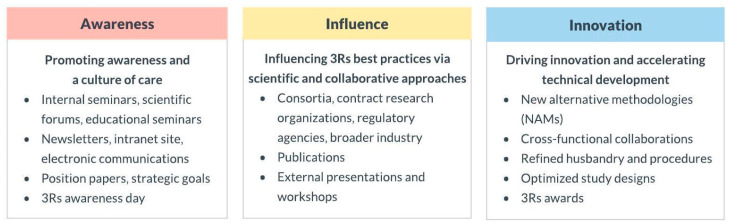
3Rs AG core focus areas (3Rs—replacement, reduction and refinement. AG—advisory group).

**Figure 3 animals-13-03863-f003:**
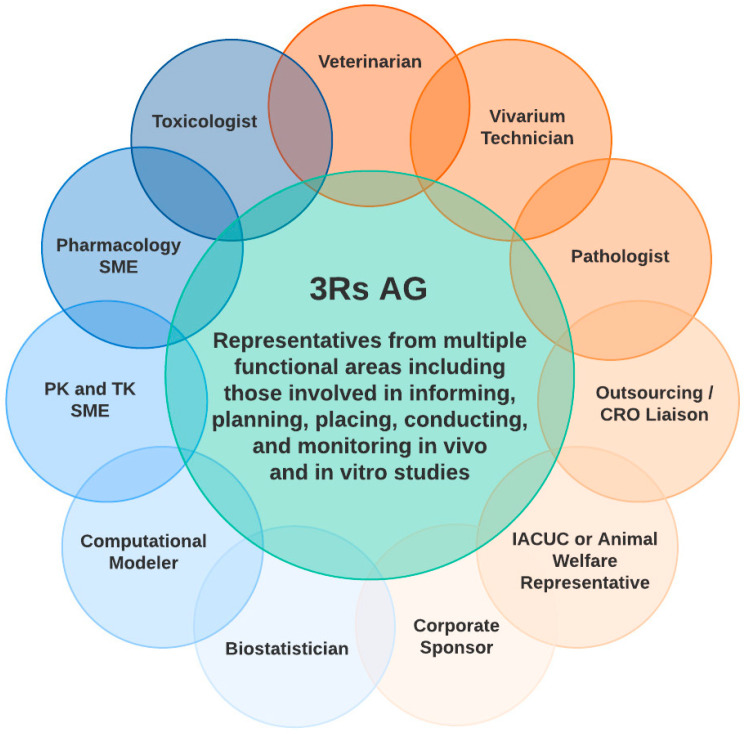
3Rs AG cross-functional engagement. Representation in a 3Rs AG is ideally multifaceted and includes complementary functions across the organization, allowing for productive conversations, collaborations, and wide-reaching impact. Key members of a 3Rs AG in the pharma sector are presented here and may vary based on the function and needs of an organization. Acronyms: 3Rs: replacement, reduction and refinement; AG: advisory group; CRO: contract research organization; IACUC: institutional animal care and use committee; PK: pharmacokinetics; SME: subject-matter expert; TK: toxicokinetics.

**Table 1 animals-13-03863-t001:** Roles and responsibilities.

Company Leadership/Sponsor	3Rs AG Leadership	3Rs AG Membership
Conveys news on internal/external events, opportunities, and issues Provides guidance to 3Rs AG leadership when requested and removes barriers to process adoption Supports the integration of practices and processes aligned with 3Rs principles Secures financial support for 3Rs awards and activities (see Section 3.1) Attends 3Rs AG outreach and educational activities (e.g., awards) Visibly supports the 3Rs AG (e.g., includes 3Rs award winners in company communications)	Maintains awareness of animal-welfare activities and issues throughout the industry and brings this information back to the AG Encourages collaborations, communication, and discussions Establishes 3Rs awards Leads AG meetings Fosters an open forum for colleagues to discuss 3Rs topics Maintains continuous engagement in communications with 3Rs membership and the greater organization (e.g., publishing a regular newsletter)	Enables cross-functional oversight of new-approach methodologies (NAMs) (in vitro, in silico, in chemico) and laboratory-animal advancements Informs the AG of external activities Encourages internal engagement Presents internally and externally on 3Rs-related topics Updates the AG on current activities, challenges being faced, solutions being evaluated/implemented, etc. Champions the advancement of 3Rs principles

3Rs—replacement, reduction and refinement. AG—advisory group.

**Table 2 animals-13-03863-t002:** Impactful 3Rs AG internal and external activities (3Rs—replacement, reduction and refinement. AG—advisory group).

Purpose	Internal	External
**Drive Innovation**	Implement an annual 3Rs award program that recognizes novel approaches that advance 3Rs principles Encourage new ways of thinking Be open to discussing and evaluating new methodologies that have the potential to reduce/replace/refine animal use	Lead and/or contribute to consortia Contribute ideas, opportunities, and feedback that advance 3Rs principles and policies at conferences/meetings Volunteer with organizations that are committed to advancing the 3Rs
**Educate and Increase Awareness**	Hold regular events (e.g., seminars) on current 3Rs topics Present posters and author communications that have wide organizational reach Voice support of the 3Rs and bring such a perspective into relevant discussions	Ask questions related to how practices/principles align with 3Rs principles Author publications that educate others on 3Rs principles Present at external meetings on 3Rs-related topics
**Foster a Culture of Care**	Encourage and foster ideas for enhanced animal welfare Review study protocols from an animal-welfare perspective Recognize teams and individuals who contribute to the advancement of 3Rs principles within the company	Encourage the enhancement of animal welfare across the industry Assess contract research organizations from an animal-welfare risk assessment perspective Increase the awareness of teams and individuals who contribute to the advancement of the 3Rs externally

**Table 3 animals-13-03863-t003:** Animal-welfare directives, regulations, and related guidance.

Directive/Regulation/Guidance	Jurisdiction/Source	Web Address
United States, Animal Welfare Act (1966)	United States	https://www.govinfo.gov/content/pkg/COMPS-10262/pdf/COMPS-10262.pdf (accessed on 29 April 2023)
United States Department of Agriculture (USDA) Animal Care, Animal Welfare Act, and Animal Welfare Regulations (The “Blue Book”) (2019)	United States	https://www.aphis.usda.gov/animal_welfare/downloads/bluebook-ac-awa.pdf (accessed on 29 April 2023)
National Research Council (NRC), Guide for the Care and Use of Laboratory Animals (“The Guide”)	United States	https://www.ncbi.nlm.nih.gov/books/NBK54050/ (accessed on 29 April 2023)
Advisory Committee to the Director (ACD) Working Group on Enhancing Rigor, Transparency, and Translatability in Animal Research (white paper)	United States	https://acd.od.nih.gov/working-groups/eprar.html (accessed on 29 April 2023)
Necessity, Use and Care of Laboratory Dogs at the United States Department of Veteran Affairs (white paper)	United States	https://nap.nationalacademies.org/catalog/25772/necessity-use-and-care-of-laboratory-dogs-at-the-us-department-of-veterans-affairs (accessed on 29 April 2023)
European Union (EU) Directive 2010/63/EU; EU Convention for the Protection of Vertebrate Animals used for Experimental and other Scientific Purposes (ETS123)	European Union	https://www.coe.int/en/web/conventions/full-list?module=treaty-detail&treatynum=123 (accessed on 29 April 2023)
United Kingdom Home Office, Animals (Scientific Procedures) Act, 1986	United Kingdom	https://www.legislation.gov.uk/ukpga/1986/14/contents (accessed on 29 April 2023)
Canadian Council on Animal Care (CCAC), Guidelines for the Care and Use of Animals in Science	Canada	https://ccac.ca/en/guidelines-and-policies/the-guidelines/ (accessed on 29 April 2023)

**Table 4 animals-13-03863-t004:** Organizations and consortia with a focus on the 3Rs (replacement, reduction and refinement).

Organization	Mission	Web Address
National Centre for the Replacement, Refinement and Reduction of Animals in Research (NC3Rs)	UK-based scientific organization dedicated to helping the worldwide research community to identify, develop, and use 3Rs technologies and approaches	https://www.nc3rs.org.uk/ (accessed on 29 April 2023)
The 3Rs Collaborative (3RsC; formerly NA3RsC)	Collaborative that aims to improve the lives of research animals, reduce their numbers while maintaining robust results, and replace them when scientifically appropriate	https://www.na3rsc.org/ (accessed on 29 April 2023)
Norecopa	Norway’s National Consensus Platform for the advancement of the 3Rs (replacement, reduction, refinement) in connection with animal studies	https://norecopa.no/ (accessed on 29 April 2023)
Center for Alternatives to Animal Testing (CAAT), Johns Hopkins, Bloomberg School of Public Health	CAAT programs seek to provide a better, safer, more humane future for people and animals	https://caat.jhsph.edu/ (accessed on 29 April 2023)
Animal Welfare Institute (AWI)	AWI fosters species-appropriate housing, compassionate care and handling, and the minimization of fear, distress, and pain for animals in research. AWI also promotes research methods that reduce the total number of animals utilized in research.	https://awionline.org/content/animals-laboratories (accessed on 29 April 2023)
Universities Federation for Animal Welfare (UFAW)	UFAW is an internationally recognized, independent, scientific, and educational animal-welfare charity concerned with improving the knowledge and understanding of animals’ needs. It promotes high standards of welfare and solutions to welfare problems for farm, companion, laboratory, and captive wild animals and those with which we interact in the wild	https://www.ufaw.org.uk/ (accessed on 29 April 2023)

**Table 5 animals-13-03863-t005:** Base-case nonclinical study design. A base-case rodent study design without mid-dose or recovery animals is used when no adverse findings are anticipated at any dose level based on previous toxicity studies. The suggested study design below can be modified to meet study objectives.

Group	Terminal	Recovery
Control	10 M/10 F	-
Low	10 M/10 F	-
Mid	-	-
High	10 M/10 F	-

## Data Availability

No new data were created or analyzed in this study. Data sharing is not applicable to this article.

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
