# Peer review of "Fostering Animal Welfare and Advancing 3Rs Principles through the Establishment of a 3Rs Advisory Group"

_animals, 2023, doi:10.3390/ani13243863_

Round 1

Reviewer 1 Report

Comments and Suggestions for Authors

Dear Authors,

Thank you very much for your work in describing a model for implementing the 3Rs in a company. I think it is a very interesting approach, maybe more of a commentary than a review.

I think it is very well explained and reasoned for each part of the work, and the figures and tables are very explanatory. 

It seems to me that on page 4, lines 114 and 147, the point would be 2.1 and 2.2 but 3 of the 3Rs has been inserted. Even so, note that both have the same title. While point 2.1 does talk about the Mission and Vision, point 2.2 talks more about mechanisms to achieve them.

In Figure 3 I would describe the acronyms. 

One point that could also be included would be rehoming. FELASA has just published recommendations; DOI: 10.1177/00236772231158863

Best regards, 

Author Response

Dear Authors,

Thank you very much for your work in describing a model for implementing the 3Rs in a company. I think it is a very interesting approach, maybe more of a commentary than a review.

Response:

The authors would like to thank the reviewer for their time and comments.  The authors have discussed this and agree that it is best suited as a Commentary. We have requested Animals change the article type to a Commentary rather than a Review.

I think it is very well explained and reasoned for each part of the work, and the figures and tables are very explanatory. 

Response: Thank you for this positive feedback!

It seems to me that on page 4, lines 114 and 147, the point would be 2.1 and 2.2 but 3 of the 3Rs has been inserted. Even so, note that both have the same title. While point 2.1 does talk about the Mission and Vision, point 2.2 talks more about mechanisms to achieve them.

Response: Thank you for pointing this out.  The section numbering has been corrected as suggested and the heading for 2.2. has been updated to “Organizational Support and 3Rs Leadership”.

In Figure 3 I would describe the acronyms. 

Response: Thank you for this suggestion.  The acronym definitions have been added to the legend of Figure 3.

One point that could also be included would be rehoming. FELASA has just published recommendations; DOI: 10.1177/00236772231158863

Response: The authors agree that this is an area where companies can have an impact.  Given the article is going to be a commentary and the authors do not have direct experience with rehoming at our company, this additional topic was not included.

Reviewer 2 Report

Comments and Suggestions for Authors

Comments

General: It is difficult to see this manuscript as an example apologetics for animal use in industrial product development and marketing. The phrase “staying ahead of the curve” in the title and no further mention of what exactly the curve constitutes within the body of the infomercial is a red flag. 

1.             The description of this manuscript as a “Review” should be changed to editorial or brief comment or most accurately, “testimonial”.

If you consider the very similar publication [Törnqvist E, et al (2014) Strategic Focus on 3R Principles Reveals Major Reductions in the Use of Animals in Pharmaceutical Toxicity Testing. PLoS ONE 9(7): e101638. https://doi.org/10.1371/journal.pone.0101638]; Google scholar indicates this paper has been cited 212 times (as of today). There is a significant wealth of scientific literature on the effect of the effect of the R&Burch-3Rs on use of animals in research.

Previously in this Journal [Hawkins, P. and Bertelsen, T., 2019. 3Rs-related and objective indicators to help assess the culture of care. Animals, 9(11), p.969.] this type of manuscript had been described as a Concept Paper.

This manuscript has 12 references and does not even reference Russell and Burch's The Principles of Humane Experimental Technique first published in 1959. It is an error to classify it as a review. 

2.             Author inflation: The manuscript has 7 authors all from the same laboratory.  The manuscript is narrative in nature without complex data to explain to the reader and little complexity to justify the numerous authors. Author inflation is a problem for the journal editorial quality more so than peer review.

3.             Science vs Management: This manuscript is focused on the management of people who work with animals not on the welfare of animals.  Citing lines 87-90 “Additionally, a 3Rs AG can serve as a hub for innovation and collaboration in strategizing and advancing study design and conduct; for example, enabling cross-functional oversight of, and collaboration on, the implementation of New Approach Methodologies (NAMs) (in vitro, in silico, in chemico)” as an unapologetic example of managerial buzzwords.

Animals as a journal has a very broad catchment in areas that are in scope. As a bureaucrat this manuscript reads much more like the latest managerial whim than a critical scientific reflection which would question if this Journal was the best fit for this manuscript.

4.             Philosophical Vagueness:  The paper flips between arguments couched in virtue ethics (Hawkins & Bertelsen, 2019) and arguments focused on business utilitarianism (line 38-39, can guide resources to generate  38 high quality study data at lower costs and with shorter timelines).  As a reader, I would refer that there was more clarity on the logic and motivation for the project, if motivations are deeper than apologetics. 

5.       Intersectionality: On line 329 is the statement “Institutional Review Board Statement: Not Applicable” reflecting the fact that the Journal has a policy of not publishing animal use research in the absence of an ethical use of animals approval process inherent in publishable scientific research. This 3r-AG initiative is in an area of human endeavour already constrained by legislation describing limits on animal use, and Institutional ethics committees approval necessary for publishing. One necessary addition is a paragraph or two that describes of how this recommended initiative intersects and supplements this hard control on animal use  (State Regulator) and additional softer control on animal use (Ethics Review & Pre-approval Process).  The use of animals in research is already the most tightly regulated use of animals in western society. It is misleading to present the 3r-AG concept as if it was stand-alone project in the already heavily institutionalized animal use in research and testing.

Author Response

General: It is difficult to see this manuscript as an example apologetics for animal use in industrial product development and marketing. The phrase “staying ahead of the curve” in the title and no further mention of what exactly the curve constitutes within the body of the infomercial is a red flag. 

  1. The description of this manuscript as a “Review” should be changed to editorial or brief comment or most accurately, “testimonial”.

If you consider the very similar publication [Törnqvist E, et al (2014) Strategic Focus on 3R Principles Reveals Major Reductions in the Use of Animals in Pharmaceutical Toxicity Testing. PLoS ONE 9(7): e101638. https://doi.org/10.1371/journal.pone.0101638]; Google scholar indicates this paper has been cited 212 times (as of today). There is a significant wealth of scientific literature on the effect of the effect of the R&Burch-3Rs on use of animals in research.

Previously in this Journal [Hawkins, P. and Bertelsen, T., 2019. 3Rs-related and objective indicators to help assess the culture of care. Animals, 9(11), p.969.] this type of manuscript had been described as a Concept Paper.

This manuscript has 12 references and does not even reference Russell and Burch's The Principles of Humane Experimental Technique first published in 1959. It is an error to classify it as a review. 

Response: The authors would like to thank the reviewer for their time and comments.

Regarding the comment on the article title, the authors are in alignment that there is no in depth evaluation on this figure of speech in the article as it was written.  The title has been changed to “Fostering Animal Welfare and Advancing 3Rs Principles through the Establishment of a 3Rs Advisory Group” which better reflects the intention and information provided within this article.

Regarding the comment on submission type:  The authors have discussed this and agree that it is best suited as a Commentary (a Concept Paper appears to no longer be an option). We have requested Animals change the article type to a Commentary rather than a Review.

 Additional references are also now integrated into the paper, including Russell and Burch’s foundational 1959 publication.

  1. Author inflation: The manuscript has 7 authors all from the same laboratory.  The manuscript is narrative in nature without complex data to explain to the reader and little complexity to justify the numerous authors. Author inflation is a problem for the journal editorial quality more so than peer review.

Response: The authors would like to thank the reviewer this comment.  Although from the same company, the authors are from differing functional areas, and have differing areas of expertise (toxicology, pathology, immunology, in vivo study design / monitoring, occupational health and safety, etc.).  This does not really come through in the author listing of company/email and therefore functional areas have been added to the affiliation section for Animals to consider.  The intention of this article is to provide insights into the value of a 3Rs AG as well as how to establish such a group based on experience at the authors’ company. Each author is an active and contributing member in the company’s 3Rs AG and contributed to authoring this manuscript.

  1. Science vs Management: This manuscript is focused on the management of people who work with animals not on the welfare of animals.  Citing lines 87-90 “Additionally, a 3Rs AG can serve as a hub for innovation and collaboration in strategizing and advancing study design and conduct; for example, enabling cross-functional oversight of, and collaboration on, the implementation of New Approach Methodologies (NAMs) (in vitro, in silico, in chemico)” as an unapologetic example of managerial buzzwords.

Animals as a journal has a very broad catchment in areas that are in scope. As a bureaucrat this manuscript reads much more like the latest managerial whim than a critical scientific reflection which would question if this Journal was the best fit for this manuscript.

Response:  This manuscript is focused on conveying the value a 3Rs AG offers to an organization.  As mentioned, lines 87-90 convey how the 3Rs AG adds value at Genentech.  To better convey that these statements are descriptive of activities and engagement in advancing 3Rs principles, additional text has been added to point the reader to the Case Studies at the end of the manuscript. 

  1. Philosophical Vagueness:  The paper flips between arguments couched in virtue ethics (Hawkins & Bertelsen, 2019) and arguments focused on business utilitarianism (line 38-39, can guide resources to generate  38 high quality study data at lower costs and with shorter timelines).  As a reader, I would refer that there was more clarity on the logic and motivation for the project, if motivations are deeper than apologetics. 

Response: The authors would like to thank the reviewer conveying this perspective. The authors agree and have adjusted this sentence (line 38-39) and removed any mention of lower costs and shorter timelines to remain aligned with a rationale based on ethics.  The authors aimed to offer perspectives on how we can achieve both without sacrificing or compromising one for the other as they can often go hand in hand.

  1. Intersectionality: On line 329 is the statement “Institutional Review Board Statement:Not Applicable” reflecting the fact that the Journal has a policy of not publishing animal use research in the absence of an ethical use of animals approval process inherent in publishable scientific research. This 3r-AG initiative is in an area of human endeavour already constrained by legislation describing limits on animal use, and Institutional ethics committees approval necessary for publishing. One necessary addition is a paragraph or two that describes of how this recommended initiative intersects and supplements this hard control on animal use  (State Regulator) and additional softer control on animal use (Ethics Review & Pre-approval Process).  The use of animals in research is already the most tightly regulated use of animals in western society. It is misleading to present the 3r-AG concept as if it was stand-alone project in the already heavily institutionalized animal use in research and testing.

Response: As this article is a commentary and no animal or human data is presented, the Institutional Review Board Statement is not required. 

Reviewer 3 Report

Comments and Suggestions for Authors

This manuscript is a plea for the establishment of 3Rs AG in biotechnology and pharmacology companies. The manuscript draws on the authors' experience at Genentech, the company where they are all employees. The text exudes much enthusiasm about their experiences, which they wish to share with a wider readership. The text is lucid, and the initiative is worth presenting to a wider audience. It would be interesting to know whether such an initiative would also be useful in academic institutions. Furthermore, it would be interesting to compare the role of the 3Rs WG with the role of ethical committees in academia and discuss their possible added value.

Specific comments

Lines 53-55: “Consideration of the 3Rs at every stage of drug development optimizes approaches and is a driving force in medical, scientific, and technological advancement [5].”

However, the 'reduction of animal use' principle in particular has been criticised as leading to underpowered studies that are difficult to replicate.

Line 60: What is the role of welfare/positive affective state in the quality of data derived from animal experiments? Please explain.

Lines 70-71: “Effective and efficient implementation of 3Rs principles is not  without its challenges (…)”.

Briefly mention some of these challenges.

Line 80ff. : 1.2. Benefits of a 3Rs AG to the Company, Employees and Industry

Shouldn't this be part of the discussion? I suggest moving the whole paragraph to the end of the manuscript, before "Conclusions".

You talk about benefits, but the reader still has no information about the 3Rs AG.

Line 101: “(…)fewer or shorter in vivo studies, (…)”.

This depends on the aims of the study. For example, longitudinal studies cannot be shortened.

Line 113, Table, column “Company leadership /  sponsor”

I'm wondering how independent a 3Rs AG is. "Provides guidance" sounds to me like "he who pays the piper calls the tune".

Line 113, Table, column “Company leadership /  sponsor”

“3Rs awards” What is the role of awards? I doubt that employees are very keen to receive an award that has no meaning or status outside the company.

Lines 153-154: “Having two co-leads in addition to a Corporate Sponsor, is an ideal 3Rs AG leadership team (…).”

Once again, how independent is 3Rs AG from corporate management?

Lines 157-159: “Having 3Rs AG branded items (e.g., t-shirts, water bottles) can also be fundamental in increasing the visibility of a company’s commitment to animal welfare and encourages a sense of pride in 3Rs AG membership.”

This may depend on the local culture. I can hardly imagine that 3Rs AG branded products would have had a positive impact in the companies/organizations where I was employed.

Figure 3: Where are the animal scientists, the behavioral pharmacologists, the animal behavior experts ....?

Line 185 (Table, column “Internal”): “Review study protocols from an animal welfare perspective”

This is a core function of ethics committees. Aren't companies required to submit animal studies to an ethics committee for approval?

Line 185 (Table, column “External”): “CROs”

Don't use the acronym, but "Contract Research Organizations", as the presumed readers of this paper will also work in academia and may not know "CRO".

Lines 199-200: “Additionally, highlighting cross-company opportunities, (…)”.

Do you mean that the contacts and discussions should extend to competing companies?

Line 264 (Table): This list is heavily weighted toward organizations in the US and English-speaking countries.

Line 265 (Table): This list is also heavily weighted toward organizations in the US and English-speaking countries. See, for example, https://doi.org/10.1177/02611929221099165

for a list of European organizations and institutes dedicated to animal welfare, 3Rs etc.

Lines 297-299: “For this reason, the 3Rs AG recommended that concurrent control groups should only be included in DRF studies for both rodents and non-rodents with appropriate justification.”

This is a recommendation that needs a lot of explanation and support by published evidence. The omission of a control group is a major departure from generally accepted study designs.

Lines 302-303: “Moreover, we also encourage using non-terminal control animals in large animal studies.”

I'm afraid I don't get the message. Please add an explanation of how and why. What is the final destination of these animals?

Line 305: “Elimination of TK satellite animals (…).”

Explain the abbreviation here (line 305). Now the explanation this abbreviation is given in lines 308-309

Line328: “Funding: This research received no external funding.”

But it was internally funded? If so, how independent were the authors in writing this paper? 

Author Response

This manuscript is a plea for the establishment of 3Rs AG in biotechnology and pharmacology companies. The manuscript draws on the authors' experience at Genentech, the company where they are all employees. The text exudes much enthusiasm about their experiences, which they wish to share with a wider readership. The text is lucid, and the initiative is worth presenting to a wider audience. It would be interesting to know whether such an initiative would also be useful in academic institutions. Furthermore, it would be interesting to compare the role of the 3Rs WG with the role of ethical committees in academia and discuss their possible added value.

Response: The authors would like to thank the reviewer for their time and comments.

Specific comments

Lines 53-55: “Consideration of the 3Rs at every stage of drug development optimizes approaches and is a driving force in medical, scientific, and technological advancement [5].”

However, the 'reduction of animal use' principle in particular has been criticised as leading to underpowered studies that are difficult to replicate.

 Response: We have added this sentence and conveyed it as a challenge in the implementation of 3Rs principles per this comment on lines 91-94 in the track changes version of the main manuscript.

Line 60: What is the role of welfare/positive affective state in the quality of data derived from animal experiments? Please explain.

 Response: The impact of chronically stressful conditions, not supporting a positive welfare state, on the physiological and psychological resilience, the allostatic state, of research animals is an underappreciated study variable. Stress stimulates the hypothalamic–pituitary–adrenal (HPA) axis, the sympathetic adrenal medullary axis, and the sympathetic and parasympathetic nerve projections that directly innervate secondary lymphoid organs (Tracey, 2009; Sternberg, 2006; Glaser, R. & Kiecolt-Glaser, J.K., 2005). Among other things, this results in the elevation of the ‘stress hormones’, cortisol, and corticosterone. Direct consequences of chronic neuroendocrine activation include deleterious effects on innate and adaptive immunity, and central nervous, cardiovascular and reproductive systems (Gurfein, B.T., Stamm, A.W., et.al., 2012;  Obernier, J.A. & Baldwin, R.L., 2006) resulting in increased variability in study endpoints, in turn necessitating use of increased numbers of animals.

References:

  • Tracey, K.J. (2009). Reflex control of immunity. Nature Reviews Immunology 9, 418–428.13
  • Sternberg, E.M. (2006). Neural regulation of innate immunity: A coordinated nonspecific host response to pathogens. Nature Reviews Immunology 6, 318–328.14
  • Glaser, R. & Kiecolt-Glaser, J.K.  (2005). Stress-induced immune dysfunction:  Implications for health. Nature Reviews Immunology 5, 243–251
  • Gurfein, B.T., Stamm, A.W., et.al. (2012). The calm mouse: An animal model of stress reduction. Molecular Medicine 18, 606–617
  • Obernier, J.A. & Baldwin, R.L. (2006). Establishing an appropriate period of acclimatization following transportation of laboratory animals. ILAR Journal 47, 364–369.

Lines 70-71: “Effective and efficient implementation of 3Rs principles is not without its challenges (…)”.

Briefly mention some of these challenges.

 Response: Added some examples of these challenges as suggested.

Line 80ff. : 1.2. Benefits of a 3Rs AG to the Company, Employees and Industry

Shouldn't this be part of the discussion? I suggest moving the whole paragraph to the end of the manuscript, before "Conclusions".

You talk about benefits, but the reader still has no information about the 3Rs AG.

 Response: Moved this section to prior to the conclusion as suggested.

Line 101: “(…)fewer or shorter in vivo studies, (…)”.

This depends on the aims of the study. For example, longitudinal studies cannot be shortened.

 Response: We agree and have removed this text.

Line 113, Table, column “Company leadership /  sponsor”

I'm wondering how independent a 3Rs AG is. "Provides guidance" sounds to me like "he who pays the piper calls the tune".

 Response: Adjusted this statement to convey “when requested” by the 3Rs AG and also added language with regards to company leadership removing barriers.  This is meant to convey that company leadership can be a resource for the 3Rs AG and help with breaking down barriers to implementation and integration of 3Rs principles.

Line 113, Table, column “Company leadership /  sponsor”

“3Rs awards” What is the role of awards? I doubt that employees are very keen to receive an award that has no meaning or status outside the company.

 Response: Added text to point the reader to Section 3.1 where the 3Rs Awards are described in greater detail.  At the authors’ company, the awards are actually regarded as prestigious and come with a monetary incentive. 

Lines 153-154: “Having two co-leads in addition to a Corporate Sponsor, is an ideal 3Rs AG leadership team (…).”

Once again, how independent is 3Rs AG from corporate management?

 Response: Added additional information on how the Corporate Sponsor functions in the 3Rs AG, which is more of a supportive role, then someone who is directing the activities.

Lines 157-159: “Having 3Rs AG branded items (e.g., t-shirts, water bottles) can also be fundamental in increasing the visibility of a company’s commitment to animal welfare and encourages a sense of pride in 3Rs AG membership.”

This may depend on the local culture. I can hardly imagine that 3Rs AG branded products would have had a positive impact in the companies/organizations where I was employed.

 Response: Added clarification that this can depend on local culture. Changing the article type to a Commentary also partially addresses the fact that experiences will vary across companies and cultures. These activities have been successful for the authors of this manuscript and understandably, may not be impactful in all organizations. 

Figure 3: Where are the animal scientists, the behavioral pharmacologists, the animal behavior experts ....?

 Response: It is possible that different organizations will have varying members on their 3Rs AG.  Given this is a Commentary based on current company experience, the membership is reflected.  The authors have noted in the figure legend that additional relevant individuals may need to be included depending on the function and needs of the organization. 

Line 185 (Table, column “Internal”): “Review study protocols from an animal welfare perspective”

This is a core function of ethics committees. Aren't companies required to submit animal studies to an ethics committee for approval?

Response: The additional review by a 3Rs AG can provide additional insights into where animal welfare can be enhanced.

Line 185 (Table, column “External”): “CROs”

Don't use the acronym, but "Contract Research Organizations", as the presumed readers of this paper will also work in academia and may not know "CRO".

 Response: Removed the acronym from this table as suggested.

Lines 199-200: “Additionally, highlighting cross-company opportunities, (…)”.

Do you mean that the contacts and discussions should extend to competing companies?

Response: Actually this is meant to convey opportunities, efforts and engagement within a company (across the company).  We have reworded to clarify. 

Line 264 (Table 3): This list is heavily weighted toward organizations in the US and English-speaking countries.

 Response: This limitation is recognized and speaks to the resources routinely utilized and sought by the authors.  In adjusting the article type to a Commentary, this will more appropriately convey the experiences of the author company rather than a global perspective.

Line 265 (Table 4): This list is also heavily weighted toward organizations in the US and English-speaking countries. See, for example, https://doi.org/10.1177/02611929221099165

for a list of European organizations and institutes dedicated to animal welfare, 3Rs etc.

 Response: This limitation is recognized and speaks to the resources routinely utilized and sought by the authors.  In adjusting the article type to a Commentary, this will more appropriately convey the experiences of the author company rather than a global perspective.

Lines 297-299: “For this reason, the 3Rs AG recommended that concurrent control groups should only be included in DRF studies for both rodents and non-rodents with appropriate justification.”

This is a recommendation that needs a lot of explanation and support by published evidence. The omission of a control group is a major departure from generally accepted study designs.

Response: Thank you for these comments. As mentioned in the manuscript, the primary objective of a non-GLP DRF toxicology study is to assess tolerability and identify the maximum tolerated dose (MTD) of a new molecular entity (NME) to help inform dosing for a GLP and relatively longer-term study. These studies rely on a dose escalation paradigm in which clinical observations, clinical pathology, and sometimes anatomic pathology endpoints are collected.  Through experience with and retrospective review of studies including controls we have determined the determination of MTD can be made without the inclusion of a comparator control group. 

Lines 302-303: “Moreover, we also encourage using non-terminal control animals in large animal studies.”

I'm afraid I don't get the message. Please add an explanation of how and why. What is the final destination of these animals?

Response: Thank you for your comment. Given the complexity surrounding this issue, we have decided to remove this sentence.

Line 305: “Elimination of TK satellite animals (…).”

Explain the abbreviation here (line 305). Now the explanation this abbreviation is given in lines 308-309

 Response: Adjusted to include the acronym definition earlier.

Line328: “Funding: This research received no external funding.”

But it was internally funded? If so, how independent were the authors in writing this paper? 

Response: Adjusted to state, “This work was funded by Genentech, Inc.”